

# Fine particles from Independence Day fireworks events: chemical characterization and source apportionment

Jie Zhang[1,] Sara Lance[1], Jeffrey M. Freedman[1], Yele Sun[2], Brian A. Crandall[1], Xiuli Wei[1,3], James J. Schwab[1]

[1]Atmospheric Sciences Research Center, University at Albany, State University of New York;

[2]State Key Laboratory of Atmospheric Boundary Layer Physics and Atmospheric Chemistry, Institute of Atmospheric Physics, Chinese Academy of Sciences, Beijing, China

[3]Anhui Institute of Optics and Fine Mechanics, Chinese Academy of Sciences, Hefei, China

*Correspondence to*:James J. Schwab (jschwab@albany.edu)

**Abstract:** To study the impact of fireworks (FW) events on air quality, aerosol particles from FW displays were measured using a High-Resolution Time-of-Flight Aerosol Mass Spectrometer (HR-ToF-AMS) and collocated instruments during the Independence Day (July 4) holiday 2017 at Albany, NY, USA. Three FW events were identified through the potassium ion ($K^+$) signals in the aerosol mass spectra. The largest FW event signal measured at two different sites was the Independence Day celebration in downtown Albany, with maximum hourly $PM_{2.5}$ of about 55 µg m$^{-3}$ at the downtown site (approximately 1 km from the FW launch location), and 33.3 µg m$^{-3}$ of non-refractory $PM_1$ at the uptown site (approximately 8 km downwind). The aerosol concentration peak measured at the uptown site occurred 2 hours later than at the downtown site. The Independence Day FW events resulted in significant increases in both organic and inorganic ($K^+$, sulfate, chloride) chemical components. Positive Matrix Factorization (PMF) of organics mass spectra identified one FW related organic aerosol factor (FW-OOA) with a highly oxidized state. The intense emission of FW particles from the Independence Day celebration contributed about 79.0% (26.1 µg m$^{-3}$) of total $PM_1$ (33.0 µg m$^{-3}$) measured at the uptown site during Independence Day FW event (07/04 23:00-07/05 02:00). Aerosol measurements and wind LiDAR measurements showed two distinct pollution sources, one from the Independence Day FW event in Albany, and the other transported from the northeast, potentially associated with another city's FW events. This study highlights the significant influence of FW burning on fine aerosol mass concentration and chemical characteristics, which is useful in quantifying the impacts of FW on air pollution, at a time when more than usual people are clustered together and breathing the outdoor air.

## 1 Introduction

Firework displays (FW) from national celebrations, such as Independence Day in United States, Spring and Lantern Festivals in China, Diwali Festival in India, Guy Fawkes Night in the UK, and worldwide New Year's Eve celebrations are known to cause short-term very high-intensity air quality degradation, especially atmospheric particle matter (PM) pollution (Seidel et al., 2015;

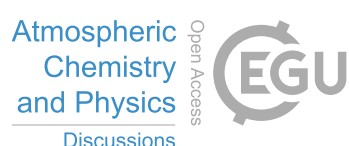

Dickerson et al., 2016; Wang et al., 2007; Jiang et al., 2015; Kong et al., 2015; Tian et al., 2014; Yang et al., 2014; Barman et al., 2008; Godri et al., 2010; Drewnick et al., 2006). Besides substantial visibility reductions lasting for hours and heavy emission of gaseous pollutants, such as nitrogen oxides and sulfur dioxide (Vecchi et al., 2008), FW events also produce large amounts of fine aerosol ($PM_{2.5}$) that are rich in sulfate, organics, potassium, and heavy metals (e.g. Cu, Ba, Al) (Moreno et al., 2007; Jiang et

al., 2015; Lin et al., 2016). The heavy metals come from the inorganic salts that are used in FW manufacturing to produce different colors, while the compounds used for oxidizing firework mixtures produce abundant potassium salts, such as potassium nitrate, potassium chlorate, and potassium perchlorate. FW produced aerosols with metal compounds were found to have greater toxicity than traffic emitted aerosols (Godri et al., 2010), and to be harmful to public health, with detrimental impacts on the respiratory system (Joly et al., 2010; WHO, 2006).

The Independence Day holiday in the United States (July 4) has been celebrated for more than two hundred years and is associated with grand FW displays around the whole country, from small towns to megacities. According to the American Pyrotechnics Association (APA) fireworks consumption figures, about 110.7 million kg of fireworks were detonated by U.S. consumers in 2016. Nationwide on Independence Day, there has been an average increase in 24-hr $PM_{2.5}$ concentration by 42% across the U.S. on Independence Day comparing to the adjacent days (Seidel et al., 2015). This indicates the importance of

studying the fine PM emission from the FW displays that are a feature of Independence Day celebrations.

Previous studies of FW fine PM were mainly based on filter methods to analyze the particle composition mass concentration (Kong et al., 2015; Yang et al., 2014), or bulk particle size distribution characteristics (Wehner et al., 2000). The long integration time for collection of filter samples (12 hours or 24 hours) loses the real-time information (composition and size distribution) of FW aerosol, and creates uncertainties in accurately quantifying these parameters (Jiang et al., 2015). The chemical composition

and size distribution of particles, as measured by more advanced aerosol mass spectrometry technology (Jayne et al., 2000; Decarlo et al., 2006, 2008; Sun et al., 2010; Sun et al., 2012), provides a new perspective on FW PM characteristics (Drewnick et al., 2006, Jiang et al., 2015). Firework particle emissions have been shown to have high potassium ($K^+$) and organic mass ratios, and the observed mass spectra of organic compounds is similar to that of secondary organic aerosol (SOA) (Jiang et al., 2015, Jimenez et al., 2009; Ng et al., 2011).

The high mass concentration and complex emission sources of background aerosol in megacities such as Beijing (Jiang et al., 2015) will to some extent mask information about the contribution of FW to ambient aerosol, and in particular the features of the organic aerosol species directly emitted from FW burning. Albany, the capital of New York State (a city with a population of

approximately 100,000 about roughly 240 km north of New York City), features relatively clean air (PM mass concentration generally below 15 µg m$^{-3}$, Zhang et al., 2018). Albany hosted a large FW display for the Independence Day holiday, which lasted for almost an hour from 22:00 to 23:00 local time (EDT: Eastern Daylight Time) on July 4, 2017 and was surrounded by smaller FW events in regional towns such as Bennington, Vermont and Saratoga Springs, New York. As will be shown, the

weather conditions during Albany's fireworks display provided a unique opportunity for a detailed investigation of the contribution of FW displays to the ambient aerosol mass concentration and chemical content, the different sources of aerosols, and the health influence of these kinds of metal-rich aerosols on surrounding areas.

In this study, aerosol particles from FW displays were measured using a High-Resolution Time-of-Flight Aerosol Mass Spectrometer (HR-ToF-AMS) and collocated instruments during the 2017 Independence Day holiday in Albany, NY. The

aerosol mass concentration, chemical composition, and size distribution are characterized. The sources of different organic aerosol, and their mass spectra character are studied. In addition, the role of transport is investigated based on LiDAR measurements and back trajectory analysis.

## 2 Methods

### 2.1 Measurement site and sampling instruments

The main ("uptown") measurement site is located at the Atmospheric Sciences Research Center (latitude: 42.7 °N, longitude: 73.8 °W, elevation: 81 m, hereafter ASRC site, as shown in Fig. S1), near the main campus of the University at Albany, which is located in the uptown (NW) section of Albany, about 8 km downwind of the Albany FW launch site (at Empire State Plaza) and near the intersection of interstates I90 and I87. The measurements of aerosol concentration were conducted from June 27 00:00 local time (EDT) to July 7 12:00, except the period from July 1 00:00 to July 2 20:00 for the maintenance of instruments. The

measurement period covered the Independence Day holiday, during which an intense Albany FW event occurred. A second ("downtown") measurement site is the Albany County Health Department measurement site (latitude: 42.63 °N, longitude: 73.75 °W, elevation: 8 m, hereafter ACHD site), which is about 1 km southeast of Empire State Plaza and about 9 km from ASRC site.

The key instrument is the HR-ToF-AMS, which was used to measure non-refractory (NR) submicron aerosol (NR-PM$_1$) chemical composition (including organics, nitrate, sulfate, ammonium, and chloride), as well as the size distributions of aerosol

chemical species (DeCarlo et al., 2006; Drewnick et al., 2005). This HR-ToF-AMS was operated under the sensitive "V-mode", and cycled through the mass spectrum (MS) mode and the particle-of-flight mode (PToF) every minute.

A TSI Scanning Mobility Particle Sizer (SMPS), which consists of an Electrostatic Classifier (EC, model 3080), a Differential

Mobility Analyzer (DMA, model 3081), and a Condensation Particle Counter (CPC, model 3785), was used to measure particle

number size distributions. Total $PM_{2.5}$ mass concentration was also obtained from an optical scattering based aerosol mass

measurement instrument (pDR-1500, Thermo Fisher Scientific Inc.), which measures the scattered light flux and uses this signal

to estimate the aerosol mass concentrations. These aerosol measurement instruments sampled from a common silica gel dryer

(RH < 37%), which was downstream from a $PM_{2.5}$ cyclone and a ~9 m long sample inlet with 10 cm inner diameter. A

supplemental airflow was continually drawn through the sample inlet, and aerosols were drawn from the centerline of this inlet

duct. Also at the ASRC site, a Leosphere Windcube 100S scanning LiDAR (hereafter LiDAR), operated from the roof of Center

for Environmental Sciences and Technology Management building (about 15 m high). The LiDAR provided high resolution (25

m range gates) wind data and backscatter profile structure. At the ACHD site, a Teledyne API Model T640 PM mass monitor

(hereafter ACHD T640) was used to provide the mass concentration of $PM_{2.5}$, and black carbon was measured by a Teledyne

API Aethalometer (model 633).

## 2.2 Meteorological parameters and back trajectory calculations

Meteorological data was obtained from the Voorheesville New York State Mesonet station (NYSM, latitude: 42.65 °N, longitude:

73.92 °N, elevation: 100 m), located approximately 8 km southwest of the ASRC site. The Mesonet site provides meteorological

data in 5-minute intervals, and includes temperature (at 2 m and 9 m), relative humidity, redundant measurements of wind

direction and wind speed, irradiance, and precipitation. On the night of the Independence Day FW event, the RH was above 90%,

with maximum wind speeds generally below 2 m/s, indicating quiescent atmospheric conditions.

Ten-hour air mass back trajectories were calculated using the NOAA ARL Hybrid Single-Particle Lagrangian Integrated

Trajectory (HYSPLIT) model, based on GDAS meteorological data (Draxler et al., 1998), and were used to study the effect of

transported FW burning aerosol on measured aerosol mass concentration at the uptown ASRC sampling site. The ending heights

of 200 m, 500 m, and 1000 m were chosen to investigate the influence of different transport layers.

## 2.3 Data analysis

The HR-ToF-AMS data were analyzed using the standard HR-AMS data analysis software – SQUIRREL v1.59D and PIKA

v1.19 (Allan et al., 2003; Canagaratna et al., 2007), to obtain the mass concentrations of different aerosol components (organics,

nitrates ($NO_3$), sulfate ($SO_4$), ammonium ($NH_4$), chloride (Cl)), and their size distributions. The default relative ionization

efficiency (RIE) values were used in the analysis (4 for ammonium, 1.1 for nitrate, 1.2 for sulfate, 1.3 for chloride, and 1.4 for




organics). The collection efficiencies (CE) used were those suggested in a previous study (Zhang et al., 2005), specifically a CE of 0.5 for inorganic compounds, and a CE of 0.7 for organic compounds based on the comparison with SMPS as shown in Fig. S2. The elemental analysis (oxygen-to-carbon (O:C) and hydrogen-to-carbon (H:C) ratio) is calculated following the improved method proposed by Canagaratna et al. (2015).

5 Due to the slow evaporation and ionization of potassium (K) (Jiang et al., 2015; Slowik et al., 2010), quantification of K mass concentrations (which are often relatively small) can be difficult and prone to large uncertainties, and potassium signals are often ignored when reporting ambient measurements. However, during FW events, K signals generally increase dramatically (Drewnick et al., 2006; Jiang et al., 2015), due to K-rich salts used in FW burning. To quantify the mass spectral signals and obtain the mass concentration of K, a RIE value needs to be specified. Previous work has used values of $RIE_k=10$ (Slowik et al., 10 2010) or $RIE_K=2.9$ (Drewnick et al., 2006; Jiang et al., 2015). In this work, an $RIE_K=2.9$ would result in the highest hourly-averaged value of K in the AMS data being as high as 36 µg m$^{-3}$, and the total HR-ToF-AMS mass concentration would be about 58.7 µg m$^{-3}$. That would be higher than the highest PM$_{2.5}$ mass concentration measured in ACHD (55 µg m$^{-3}$), a result that is not expected considering that HR-ToF-AMS only measures NR-PM$_1$ species. Furthermore, a $RIE_K = 2.9$ would result in a K/S (potassium to sulfur) ratio as high as 16, which deviates greatly from the value of 2.75 suggested by Drewnick et al (2006).

15 A $RIE_k=10$ results in lower K mass concentration by a factor of 3, and the resulting maximum for K hourly-averaged mass concentration is 10.6 µg m$^{-3}$, with the corresponding highest HR-ToF-AMS total aerosol mass concentration equal to 33.3 µg m$^{-3}$, substantially lower than the ACHD PM$_{2.5}$ mass concentration. For this RIE, the K/S ratio works out to be 5.3, nearer to the expected 2.75, and the K mass fraction during the maximum FW burning period is about 33.5%, which is consistent with the 30% result of Drewnick et al (2006). Therefore, an $RIE_K=10$ for this study is reasonable. However, due to a lack of formal 20 analysis about $RIE_k$ formulations, large uncertainties are still possible for the quantification of the mass concentration of K. For the isotopic $^{41}K^+$, a ratio of 0.0722 was used, and the total K mass concentration in the following section is inferred to be the combined mass concentration of $K^+$ and $^{41}K^+$ (Jiang et al., 2015).

The HR-ToF-AMS organic mass spectra were analyzed using positive matrix factorization (PMF) (Paatero and Tapper, 1994) to resolve different organic aerosol factors, and the solutions were evaluated using the PMF Evaluation Tool (PET, v2.08D, Ulbrich 25 et al., 2009). In this study, a five-factor solution with f$_{peak}$=0.2 (Q/Q$_{expected}$=3.9, Fig. S3) was chosen as the optimal result, based on the evaluation of spectral profiles, diurnal variation, and correlations with external tracers (Fig. S4) (Zhang et al., 2011).





## 3 Result and discussion

The time series of aerosol hourly-averaged mass concentration measured by all instruments exhibited the same general behavior during the measurement period (Fig. 1a). From June 29 12:00 until late in the day of June 30, the aerosol mass concentration displayed an increasing trend which also corresponded to high RH conditions (generally > 75%, Fig. 1b), likely due in part to

hygroscopic aerosol growth and liquid-phase organic formation (Sun et al., 2011). This is supported by the steady increase in the aerosol median diameter and the growth trend shown in the aerosol number size distribution (Fig. S5). A dramatic decrease in aerosol concentration occurred at June 30 14:00, due to scavenging by rain (Fig. 1b).

From the overnight hours of July 2 onward, there were several aerosol concentration peaks at nighttime, and the biggest one was during the night of July 4, with maximum recorded concentration of 55 $\mu$g m$^{-3}$ for ACHD T640 PM$_{2.5}$ data and 33.3 $\mu$g m$^{-3}$ for

ASRC HR-ToF-AMS NR-PM$_1$, which temporally extended into the early morning of July 5. These PM peaks are hypothesized to be dominated by FW events, which are readily identifiable using the K signal. There are two reasons why the HR-ToF-AMS observed lower PM concentrations than the ACHD T640. One reason is that HR-ToF-AMS only measures non-refractory aerosol species below 1 $\mu$m (DeCarlo et al., 2006). The second is due to diffusion and dispersal of the aerosol plume during the transport from the source location of the FW burning to the uptown ASRC site. A time difference of about two hours between the peaks of

the ACHD T640 data and ASRC instruments was observed, indicating the transport time of aerosol from the launch site downtown to the ASRC measurements site, and providing for an advection velocity of about 1 m s$^{-1}$. Excluding the high points during the night of July 4 (the FW event period), the high coefficient of determination (R$^2$=0.72) between ACHD T640 and ASRC HR-ToF-AMS suggests generally similar conditions - typical of regional aerosol pollution processes during normal non-FW days.

### 3.1 Identification of firework events and aerosol composition

Among the PM peaks, three were identified as significant FW events based on the prominent potassium signal peaks (Jiang et al., 2015; Drewnick et al., 2006). As shown in Fig. 2a, these three K peaks occurred during the nighttime hours of July 2, July 3, and July 4, with the highest peak occurring on the night of July 4. Observed concentrations of K were as high as 11.33 $\mu$g m$^{-3}$ (July 5 00:00 EDT), which is about 63 times of the background value (0.18 $\mu$g m$^{-3}$, averaged July 4 09:00 to 17:00 EDT). The high K

signal was present until about 09:00 EDT the following morning. Based on the time variation of K, we define the Independence Day FW event time period as July 4 23:00 to July 5 02:00 EDT. Apart from these three prominent peaks, there were still some spikes of K during the measurement period, likely related to small-scale, localized, or more distant firework burning.





During the Independence Day FW event period (July 4 23:00 to July 5 02:00 EDT as defined above), the averaged mass concentrations of total aerosol, organics, K, $SO_4$, $NO_3$, and Cl were all clearly elevated compared to earlier that day (July 4 9:00 to 17:00 EDT), increasing from 6.0 µg m$^{-3}$ to 27.2 µg m$^{-3}$ (4.5 times), 5.0 µg m$^{-3}$ to 12.1 µg m$^{-3}$ (2.4 times), 0.2 µg m$^{-3}$ to 9.1 µg m$^{-3}$ (45.5 times), 0.5 µg m$^{-3}$ to 4.1 µg m$^{-3}$ (8.2 times), 0.09 µg m$^{-3}$ to 0.8 µg m$^{-3}$ (8.8 times), and 0.01 µg m$^{-3}$ to 0.84 µg m$^{-3}$ (84

times) respectively (Fig. 2a). These enhancements are similar to the results of the Drewnick et al. (2006) study. Absolute concentrations of chloride from the FW burning were relatively low (compared to potassium) due to the semi-refractory character of metal chlorides (Drewnick et al., 2006). Still, the rise in chloride concentration was dramatic, with values nearly 84 times greater than the daily averaged value. The observed FW events had only minor effects on $NH_4$, as in the Drewnick et al. (2006) study, due to a lack of $NH_4$-containing material in the fireworks, and also indicates the increased $SO_4$ and $NO_3$ came from

K-rich salt instead of $(NH_4)_2SO_4$ and $NH_4NO_3$. The time series shows that organics and $NO_3$ sustained broader peaks than K and $SO_4$, possibly due to contribution of the vehicular traffic emission to organics and $NO_3$ and the nighttime $NO_3$ formation (Sun et al., 2011; Xu et al., 2015).

Before and after these three FW events (before July 1 00:00 and after July 5 12:00 EDT), the mass fraction of the HR-ToF-AMS chemical components was fairly stable, with organic compounds constituting the major fraction of NR-PM$_1$, with 70.5% (before

July 1) and 81.1% (after July 5) on average, followed by $SO_4$ (16.4% and 8.5% respectively, Fig. 2b and Fig. S6). The stable component ratio indicates relatively constant PM$_1$ sources in and near Albany, suggesting a stable aerosol evolution process during these two time periods. A higher ratio of organics after FW events may be the result of the primary emissions of organics during the FW events, or the formation of new secondary organic aerosol (Li et al., 2013; Kong et al., 2015, Wang et al., 2007). During the FW events, and especially during the Independence Day FW event, most aerosol components clearly increased, and

there was a large difference in the mass fractions of aerosol components compared to before and after periods, highlighting the influence of FW burning. The indicator of FW events, K, displayed large increases during the FW events, contributing roughly 33% to the ambient aerosol during the four-hour FW event period, and up to 45 % of ambient aerosol at the peak hour of July 4 23:00 EDT.

Most of the Independence Day FW PM$_1$ aerosol located at the size range of 200 to 500 nm (volume mobility diameter, Fig. 3a),

and FW-averaged size distribution of NR-PM$_1$ aerosol chemical components (Fig. 3b) showed externally mixed characteristics, with sulfate and nitrate peaking at ~550 nm (D$_{va}$: vacuum aerodynamic diameter), organics peaking at ~400 nm (D$_{va}$). The size distribution of K is complicated due to poorly characterized surface ionization properties of the evaporated K-containing species and is not presented here. Different peaks for the different chemical components suggest a non-uniform mixing state for organics,



and sulfate/nitrate, possibly related to different kinds of fresh emitted aerosols from FW. As discussed above, the $SO_4$ and $NO_3$ were likely mainly from K-rich inorganic salt, such as $K_2SO_4$ or $KNO_3$, and organics are from burning organic materials or reaction with FW oxidizer, which will be discussed in next section.

### 3.2 Source apportionment

### 3.2.1 PMF results

Based on the evaluation of spectral profiles, diurnal variation, and correlations with external tracers, five organic aerosol (OA) components, including hydrocarbon-like OA (HOA), a biomass burning OA (BBOA), a semi-volatile oxygenated OA (SV-OOA), and two low-volatility oxygenated OAs (LV-OOAs, one is named as FW-OOA and another one is named as LV-OOA) are identified, as shown in Fig. 4 and Fig. S3. HOA and BBOA are the two smallest components in this data set. The

HOA is characterized by hydrocarbon-like ions ($C_xH_y^+$ family), with evident morning and evening rush hour peaks. The BBOA is characterized by prominent signals at m/z 60 ($C_2H_4O_2^+$) and m/z 73 ($C_3H_5O_2^+$), with increasing concentrations during evening hours. The sum of HOA and BBOA had similar variation in its time series as the BC data from ADHC (Fig. S4a) with a moderate $R^2$ (0.38). These two components are mainly identified with vehicular traffic and wood-fire emissions, such as bonfires, barbeques, or other small-scale celebrations. The SV-OOA spectral profile is characterized by a O:C ratio of roughly 0.6, and a

high $f_{43}/f_{44}$ ratio of ~1 (Ng et al., 2010). SV-OOA also exhibited daily maxima during mid-afternoon, and it was moderately correlated with concurrent $NO_3$ before July 1 00:00 EDT, with a $R^2 = 0.48$, giving some indication of the local source characteristics of SV-OOA formation (Zhou et al., 2015).

The mass spectra of the two LV-OOAs were similar in this study, as shown in Fig. S3c, but they demonstrated different time series behaviour (Fig. S3d), indicating the presence of different types of OOAs. In comparison to LV-OOA, FW-OOA contains a

higher H:C ratio (1.33 vs. 1.20), and higher signals at *m/z* 29 (CHO$^+$: 0.05 vs. 0.01), m/z 60 ($C_2H_4O_2^+$: 0.006 vs. 0.0004) and m/z 73 ( $C_3H_5O_2^+$: 0.004 vs. 0.0008), as shown in Fig. 5. The FW-OOA is believed to be related to the organic aerosol released by FW due to similar variation trends in the K signal (Fig. S4c), with a high coefficient of determination ($R^2$=0.70). Compared to previous AMS fireworks papers, this is the first time that this form of organic aerosol has been separated from overall FW burning emission and contributed ~51.2% (6.2 µg m$^{-3}$) of the organic aerosol (12.1 µg m$^{-3}$) during the Independence Day FW

event period. FW-OOA is likely directly emitted from FW (Jiang et al., 2014, Drewnick et al, 2006), and produced by the chemical reaction of binding agents, such as dextrin, with the oxidizers.



The LV-OOA shows a more traditional spectral character comparable to those seen in earlier studies. It has a high O:C ratio (0.84), low $f_{43}/f_{44}$ ratio (0.25), and a high coefficient of determination ($R^2$=0.86) with concurrent $SO_4$ before July 1 00:00 EDT. This is thought to be related to highly oxidized secondary organic aerosol (Ng et al., 2010). Here we only considered the time period before the FW events to calculate correlation coefficients, since the $SO_4$ and $NO_3$ from burning fireworks will change the

basic relationship with background organics. In this study, SV-OOA showed a high coefficient of determination ($R^2$=0.94) with LV-OOA before FW events (Fig. S4d), indicating that the bulk of the LV-OOA at Albany was likely produced locally through oxidation of SV-OOA during the measurement period, instead of being carried in via long-distance transport.

### 3.2.2 Mass closure and source apportionment from FW events (July 4th)

Assuming that the aerosol is charge balanced and that the ionic species can be identified or assumed from the mass spectra

(Wang et al., 2016), the different inorganic salts can be estimated using ion-balance considerations. $NH_4$ is apportioned first into $(NH_4)_2SO_4$ and then into $NH_4NO_3$, based on the equivalent ratio of $SO_4$ to $NO_3$. Cl is ignored in this context due to its very low concentration before FW events.. The residual $SO_4$, (calculated by subtracting $SO_4$-in-$(NH_4)_2SO_4$ from total $SO_4$), the residual $NO_3$, and Cl are combined with measured K to form $K_2SO_4$, $KNO_3$, and KCl, which are used to oxidize firework mixtures. These K compounds would be assumed to comprise inorganic K, while the residual K, calculated by subtracting inorganic K from total

K, would be treated as organopotassium. Organopotassium is strongly correlated with FW-OOA, as shown in Fig. S4c. The density for each inorganic salt was averaged together, and assuming the density of organics factors (including those containing organopotassium) is 1.2 g $m^{-3}$, the density of the ambient aerosol bulk was then estimated to be mostly in the range 1.3 – 1.5 g $m^{-3}$, as shown in Fig. S2. Figure S2 also shows that HR-ToF-AMS and SMPS determined mass concentrations exhibit similar behavior, except that the mass concentration measured by the SMPS was as much as 12 μg $m^{-3}$ larger than that from the AMS

during the Independence Day FW event. This may be caused by the uncertainty of the RIE for K, the semi-refractory character of other inorganic metal salts, and/or the exiting of BC producing from the black powder explosives (Drewnick et al., 2015; Wang et al., 2016).

Figure 6 shows the time series of each inorganic salt from the ion balance calculation, as well as the different OA components. Here, OrgK is the sum of Organopotassium and FW-OOA, OOA is the sum of LV-OOA and SV-OOA, and "others" indicates

the difference between SMPS and HR-ToF-AMS (the aerosols not measured by HR-ToF-AMS). OOA, $(NH_4)_2SO_4$, and $NH_4NO_3$ showed no significant trends in concentration from July 4 12:00 to the end of the period of study, indicating that these species comprised the local background concentration, and contributing about 13.8% of total PM during the FW event period (07/04 23:00-07/05 02:00). BBOA+HOA exhibited consistently elevated concentrations from the night of July 4 to early morning of



July 5, likely be related to traffic emissions, wood, and charcoal smoke, a result that is supported by the large numbers of smaller particles emitted before and after FW event. BBOA+HOA contributed a relatively small amount, about 7.2% of the total PM. The biggest contribution came from FW burning, as inorganic and organic K-rich aerosol (including "others" compounds) comprised 79% of the total (26.1 μg m$^{-3}$ of 33.0 μg m$^{-3}$). Assuming this ratio did not change during transport from the launch site, the metal-rich aerosol increases caused by the FW burning in downtown Albany (the ACHD site) would have reached a averaged value as high as 43.0 μg m$^{-3}$ (07/04 21:00-07/04 23:00), higher than the NAAQS PM2.5 24-hr threshold value of 35 μg m$^{-3}$, and considered harmful to public health. After the FW display, the high LV-OOA mass concentration and low SO$_4$/NO$_3$ mass concentration persisted to the end of the period of study, with an increased f44 ratio and decreasing f43 ratio (indicators of aged aerosol, as shown in Fig. S7). This implies that FW-OOA may be converted into LV-OOA in the ambient atmosphere, and demonstrates the persistent effects of FW burning (Kong et al., 2014).

### 3.3 Pollution processing and meteorological conditions

The wind LiDAR on the roof of the ASRC uptown site reports data as the carrier-to-noise ratio (CNR) and wind direction, and this data is used to describe the aerosol transport and mixing process (Aitken et al., 2012). The LiDAR data for the Independence Day FW event is shown in Fig. 7. During the late evening on Independence Day (July 4, 20:00- 23:00 EDT, July 5 00:00-02:00 UTC), the winds above the ASRC sampling site were weak easterlies, and from the late night onward (July 4 23:00 local time, Jul 5 03:00 UTC) there was a temporal maximum in low-level CNR as shown in Fig. 7, concurrent with a high-concentration aerosol cluster over the site at about 400 meters above observation level, as indicated by relatively cool colors representing higher CNR values. This burst of high CNR values is thought to result from the high-altitude FW aerosol transport from the celebrations in downtown Albany, east of the site. In the following half hour, a lower altitude aerosol cluster occurred above the ASRC sampling site, coincident with the increases in aerosol mass concentrations measured by the ASRC instruments. This lower-altitude aerosol cluster may have been caused by subsidence or diffusion, which is the mixing the high-altitude aerosol with the lower atmosphere, or it may have been caused by the aerosol emission of low-altitude FW transported westward from Empire State Plaza. The aerosol cluster lingered over the ASRC site for almost ten hours, matching up with the persistent and slowly decaying aerosol mass concentrations measured by the other instruments. At about 07:00 EDT (11 UTC) on July 5, there was another high-altitude (1000 m) and near-ground aerosol cluster that passed over the ASRC site, producing a second aerosol peak measured on-site. It is quite plausible that this is related to transport of FW aerosol from other areas. A 10-hour back trajectory shows the 500 m wind passing over the town of Bennington, Vermont (Fig. S8), which had its own FW celebration on the evening of July 4. From July 4 23:00 – Jul 5 08:00 EDT, the lower surface layer wind speed was low, suggesting that the



aerosol cluster was near static, and about 12 hours was required (Jul 4 23:00 – July 5 11:00 EDT) to return to normal aerosol concentration levels (~6 µg m$^{-3}$). After July 5 09:00 EDT (13 UTC), the aerosol layer was again elevated, impacted by the increasing planetary boundary layer height, and ASRC instruments recorded rapid reductions in aerosol mass concentration. The firework emissions, which occurred in an environment with a static planetary boundary layer, had combined with the transported

FW aerosol from other celebrations in the region and resulted in high aerosol mass concentrations over the Albany area for several hours, lasting through the night.

**4 Conclusion**

Chemical characterization and source apportionment of the submicron aerosols from FW displays during the 2017 Independence Day holiday are quantified here for the city of Albany, New York. The hourly maximum mass concentration of NR-PM$_1$

measured using the ASRC HR-ToF-AMS was 33.3 µg m$^{-3}$, almost 5 times the daytime background mass concentration, while PM2.5 at the downtown ACHD site was ~67% higher (55 ug m$^{-3}$). There were significantly elevated aerosol chemical component mass concentrations (organics, K, sulfate, chloride) as a result of the Independence Day FW display, especially K, which spiked from 0.2 µg m$^{-3}$ to 9.1 µg m$^{-3}$, signifying the influence of burning fireworks. The size distribution of aerosol compounds showed externally mixed characteristics, with sulfate and nitrate peaking at ~550 nm (Dva), organics peaking at

~400 nm (Dva). PMF analysis revealed unique oxidized organic aerosol compounds (FW-OOA), which showed a high correlation (R$^2$=0.8) with the variations in K. FW-OOA shows a higher ratio at m/z 60 and m/z 73, and a lower m/z 43 and m/z 44 than the general high oxidized secondary organic aerosol does, and contributed about 51.2% (6.2 µg m$^{-3}$ of 12.1 µg m$^{-3}$) of total organics during FW events. Our study shows that the total contribution from fireworks to the total aerosol concentration could be as high as 79% (26.1 µg m$^{-3}$ of 33.3 µg m$^{-3}$) based on source apportionment calculations, while congested vehicular

traffic and wood-burning for smaller events like such as bonfires and barbeques contribute another 7.2% during the Independence Day FW event (07/04 23:00-07/05 02:00 EDT). The morning following the FW display, another aerosol mass peak occurred, which is likely the transport of FW aerosol from other towns, most likely from Bennington, Vermont. Under this kind of weak advective mixing, during the first several hours after FW displays, enhanced aerosols with rich metal materials could be very harmful to the nearby area residents, especially to the people nearest the display area and living in the downwind

direction, and this condition could be more severe for areas of high population density.



Acknowledgements. This work has been supported by the New York State Energy Research and Development Authority (NYSERDA) contract number 48971. Special thanks go to New York State Department of Health for providing data from the ACHD site.

Competing interests. The authors declare that they have no conflict of interest.





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





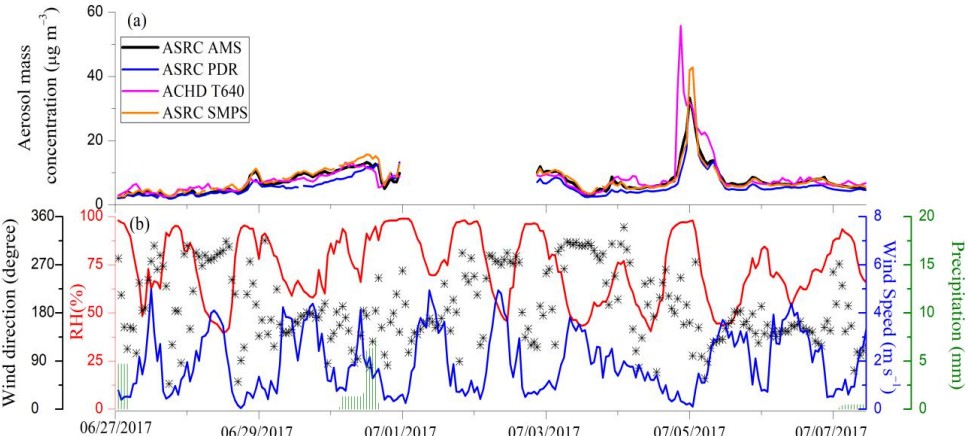

**Figure 1. Time series of (a) aerosol hourly-averaged mass concentration measured by ASRC AMS (PM₁), DRX (PM₂.₅), PDR (PM₂.₅), and ACHD T640 (PM₂.₅) (μg m⁻³); (b) NYSM meteorological parameters, with relative humidity (%), wind direction, wind speed (m/s), and precipitation (mm)**

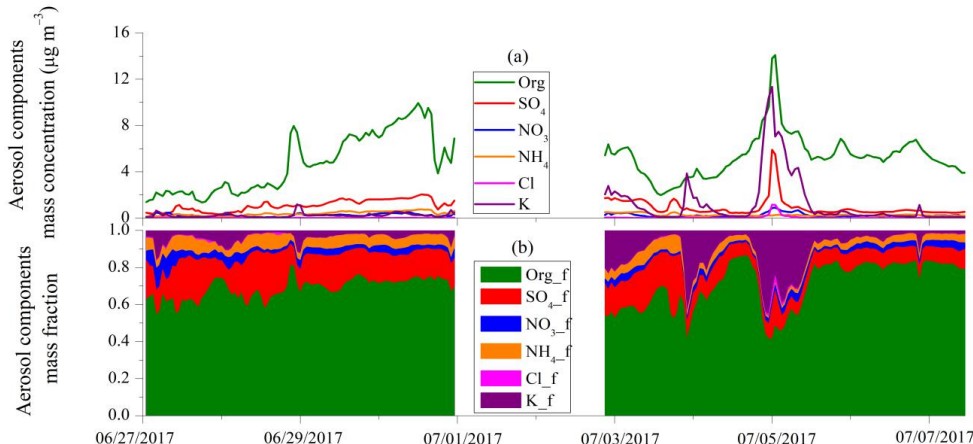

**Figure 2. The time series of (a) the mass concentrations of aerosol component (Organic, SO₄, NO₃, NH₄, Chl, and K) from AMS; and (b) the mass fraction of each component.**


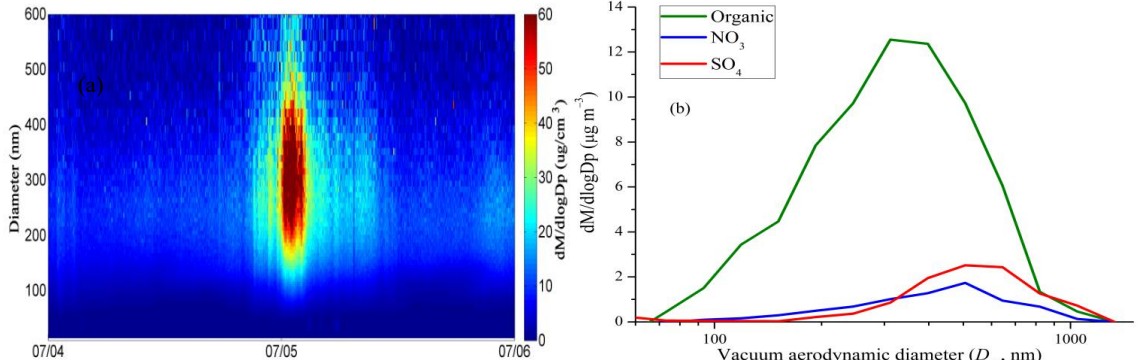

**Figure 3. (a): Mass size distribution (mobility diameter, density is set to be 1 g cm⁻³ here) measured by SMPS from July 4 12:00 to July 6 12:00; (b) AMS measured size distribution (vacuum aerodynamic diameter) for SO₄, organics and K compounds from July 4 12:00 to July 6 12:00**

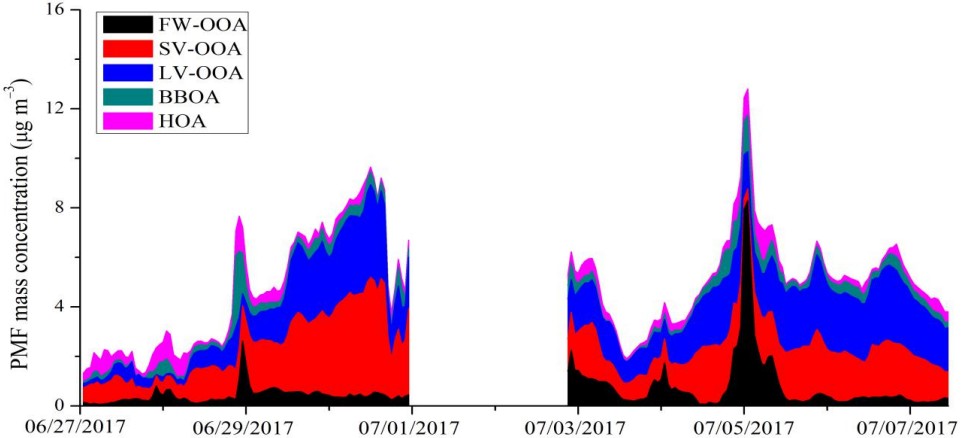

**Figure 4. Time series of the five organic aerosol factors determined from PMF analysis.**



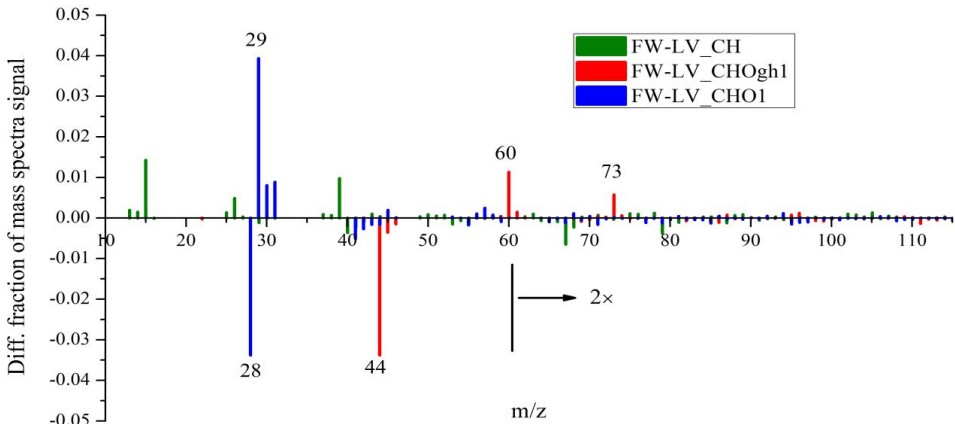

**Figure 5. The difference of the mass spectra of FW-OOA and LV-OOA. Here, both FW-OOA and LV-OOA mass spectra**

**signals were normalized to be 1.**

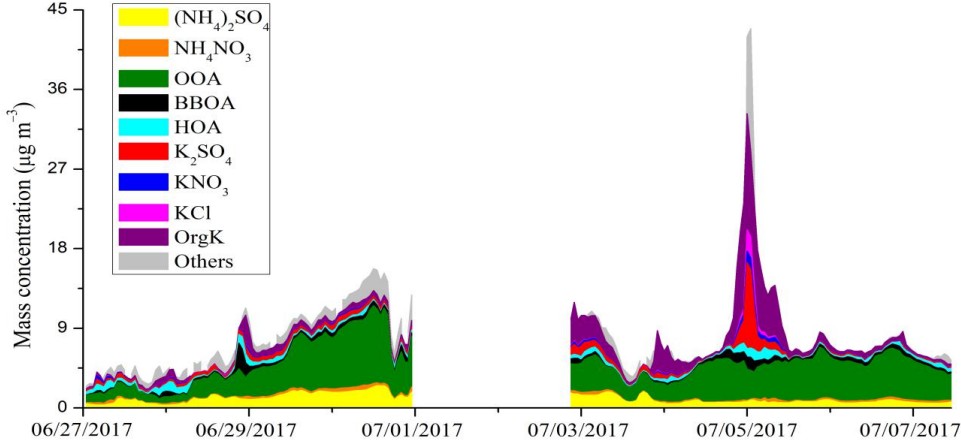

**Figure 6. Time series of different kinds of inorganic salt and organic compounds estimated based on the ion-balance**

**calculation.**





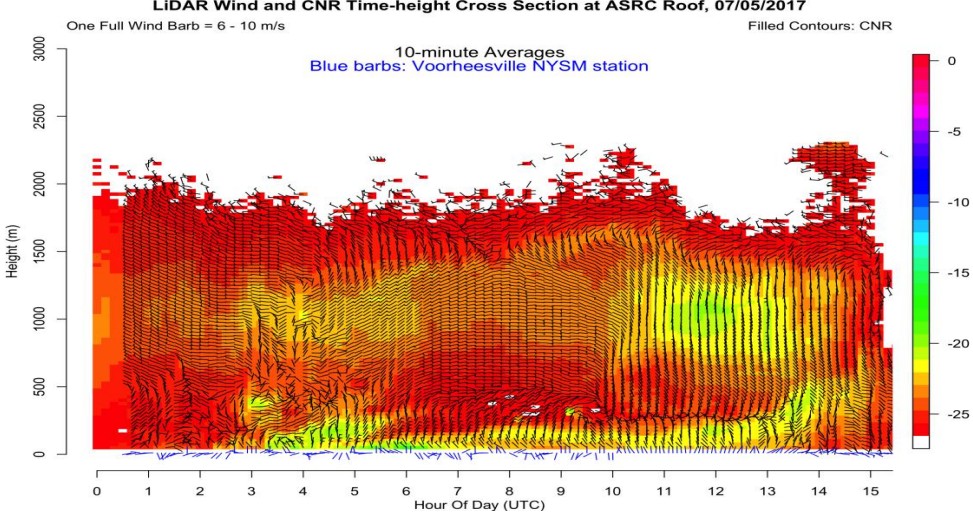

**Figure 7. Time series of LiDAR data on Independence Day (to match the wind back trajectory time in Fig.8, UTC time**

**was used here)**