# Peer review of "Fine particles from Independence Day fireworks events: chemical characterization and source apportionment"

_Atmospheric Chemistry and Physics, 2018_

## Referee Comment (RC1) · Anonymous Referee #1 · 22 Jun 2018

In their manuscript "Fine particles from Independence Day fireworks events: chemical characterization and source apportionment" Zhang et al. present results from aerosol measurements during and around the Independence Day fireworks in Albany, NY. Aerosol composition and species-resolved size distributions were measured using a HR-ToF-AMS. In addition, several instruments were measuring particle size distribution, particle mass concentration and black carbon concentration. Finally, a LIDAR was applied to determine backscatter and wind information. The authors present an analysis of the chemical composition of the fireworks aerosol as well as that of the aerosol during other times, not affected by the fireworks as determined by AMS. This includes a PMF analysis of the AMS organics. This analysis suggests that there have been

other, less intense, events which were affected by fireworks activities during the days preceding Independence Day. In addition, PM concentrations during and outside the fireworks event were compared and the relative and absolute contribution of the fireworks to the ambient aerosol are discussed. Finally, LIDAR measurements were used to investigate transport of aerosols from fireworks activities both from Albany and from the nearby Town of Bennington. The manuscript is well and clearly written. The analysis of the data makes the impression that it was performed thoroughly. However, when reading the manuscript I found a couple of inconsistencies, very different depth of information for different issues, or unclear presentation of results (see detailed comments below). Furthermore, I think the presented analysis is well behind what could be done with this interesting data set and thus neglects a significant fraction of its potential. So far the authors present a pure description of some observations during an individual event with limited general importance. In order to qualify for ACP and to provide results that can be transferred to other, similar situations, it would be desirable to extend the data analysis beyond the current state. Some examples are: • Comparison of the size distributions (total and species-resolved) measured during the Albany fireworks event, during the arrival of the Bennington fireworks aerosol and during the times not affected by fireworks could provide interesting information on the characteristics and temporal evolution of fireworks aerosols. • Comparison of the organics mass spectra of the fireworks aerosol during the Albany fireworks event and during arrival of the Bennington fireworks could provide information on aging of fireworks aerosol. For this purpose the fireworks contribution to the organics needs to be determined in a different way, e.g. by separate PMF analysis of both time intervals or by determining the difference between event and non-event times, instead of using a constant PMF factor. • A more general determination of the contribution of fireworks aerosol to the aerosol burden at a city like Albany, including an estimate on how large the contribution within the affected days or the full year is, in comparison to contributions from other sources would enable the reader to better judge the relevance of such events. This could also include an estimate of the indirect contribution of such an event due to enhanced traffic

emissions around the event.

Detailed comments:

Several fireworks events were identified during the measurement period, based on the appearance of the potassium ion in the aerosol mass spectra. I think the authors should explain the criteria more specific here since the potassium ion is also observed with the AMS when measuring biomass burning aerosols. It would be interesting whether during the observation of fireworks events with the AMS also other ions, which could be from the fireworks, were observed, e.g. Sr, Na or other metal ions.

I have a general problem with the description of the measurement locations. According to Figure 1, the wind direction was approximately 180°-270° during the main FW event. According to Figure S1 the ASRC site was located about NW of the fireworks site, i.e. it needs wind from approx. 100°-130° to transport the fireworks aerosol to the measurement site. The ACHD site was located approx. to the SE of the fireworks site, i.e. it needs wind from approx. 280°-320° to transport fireworks aerosol to the measurement site. While for the relatively close ACHD site general dispersion of the fireworks aerosol in the area would probably also transport aerosols to the measurement site, this seems unlikely for the ASRC site, which is almost in the opposite direction compared to the needed wind direction. How can you explain the transport of aerosol from Empire State Plaza to the ASCR site under these meteorological conditions?

I find the presentation of time series of aerosol concentrations (e.g. in Figs. 2, 4, and 6) stacked on top of each other not very helpful when one tries to see temporal variability of any of the mass concentrations. Generally, temporal variability is seen well only for the lowermost trace and in addition – for graphs which show fractional contributions like Fig. 2 – for the uppermost trace. For all other traces only the largest changes can be identified in this type of graph.

The presentation of all the aerosol data for a 10.5 days interval makes it hard to identify the detailed features during the main fireworks event which were discussed within the

text. It would be very helpful for the reader to have graphs which show only the time interval around the fireworks event. In addition, it would be very helpful if Figure 7 would show the same time interval as those figures and if the time axes would agree (i.e. not using UTC for one graph and local time for the others).

P2L4: Fireworks aerosol is mainly found in PM1, not in PM2.5, as shown in this manuscript as well as e.g. in the Drewnick et al., 2006 paper.

P2L11-15: Here consumption figures of fireworks in the US are given. It would be interesting to also have numbers for July 4 and may be for the Independence Day fireworks in Albany. The 42% increase of 24-hour PM2.5 – how much is this in absolute concentrations?

P2L21: Here two Sun et al. papers are cited. The Sun et al., 2010 paper is not in the reference list. Both papers do not seem to be adequate references for the general measurement capabilities of advanced aerosol mass spectrometry technology like the Jayne et al. and the DeCarlo et al. papers.

P3L4-7: Here it seems that some statements are exaggerating. Why was the meteorological situation such that it provided a "unique opportunity" for investigation of the contribution of FW displays to ambient aerosol concentrations? As stated above, the wind direction seemed to be even very unfavorable. Also I do not see how the "health influence" of FW-related aerosols was investigated in this study.

P3L16 and L22 and P4L15: It would be helpful if the authors would specify above what the elevation was measured. I guess in L16 it is a.s.l. and in L22 it is above ground, on P4L15 it could be both. Furthermore, if here and in the following paragraph some detailed information is provided, it would be nice if this would be done more consequently: Why is the sampling height provided for the ACHD site but not for the ASRC site? Where was the AMS instrument located, where the SMPS and the pDR-1500? Why is the sampling line length for the inlet duct at ASRC (?) provided, but not the length of the sampling line from the duct to the instruments and also not for the inlet

at the ACHD site? For estimate of inlet line losses, also the diameters of the lines and their orientation would be needed.

P3L19: According to the statement here, the gap in the measurement data was due to maintenance of the instruments. To me it seems very strange that maintenance started at midnight, that maintenance lasted for 44 hours, and that all instruments have exactly the same downtime.

P5L5 and the whole paragraph: "... slow evaporation and ionization of potassium ...". "evaporation" should be "vaporization". While potassium might vaporize slowly, ionization is probably not slow. In addition to this effect, the quantification of potassium in the AMS is also difficult due to the fact that potassium can be surface-ionized at the vaporizer surface during the vaporization process. This effect will directly affect RIE of potassium. Since this effect – and as a consequence the RIE of potassium – strongly depends on the vaporizer temperature, the history of the vaporizer and the tuning of the AMS, it makes little sense to use RIE values for potassium obtained in measurements with other AMS but it would be better to determine the own RIE value.

P6L2: The statement that "all instruments exhibited the same general behavior" seems a bit exaggerated. There are sometimes strong differences between the various PM2.5 measurements up to more than a factor of two. There are also sometimes strong differences in the trends.

P6L3-5: I also do not agree with the explanation of the increasing trend in aerosol concentrations with hygroscopic growth. An increasing trend is observed from 6/27 until 6/29 according to Figure 1. During this time RH values varied strongly and no correspondence of the increase in PM and RH can be observed for this whole time interval. Furthermore, hygroscopic growth of particles should not affect AMS measurements of mass concentrations and also not measurements performed with a dryer before the instrument.

P6L8-11: The aerosol concentration peaks during the nights of July 2-3 were also

identified as FW-related due to their high K signal. Is there any other evidence that these peaks are actually from FW – e.g. similar K/S ratio as observed during the July 4 peak? Was there fireworks reported or observed in the area around the measurement sites? According to the magnitude of these additional FW peaks either a massive fireworks event in the order of 30% of the July 4 fireworks or fireworks which were burned very close to the AMS site would be needed to generate these peaks. Is this realistic?

P7L3-5: It is very hard to assign all these numbers to the respective species.

P7L8-10: If SO4 and NO3 are from K-rich salts and if slow vaporization of K results in under-estimation of K, this should also result in under-estimation of the related SO4 and NO3.

P7L10-11: I do not agree with the statement here. According to Figure 2, organics and K show broader peaks than SO4. If all three are from the same source (fireworks), is this difference a sign of chemical processes occurring with the FW aerosol?

P7L13-15: If SO4 fraction varies between 8.5 and 16.4% (a factor of two), I would not call this "fairly stable". As a consequence I would not say that this indicates constant sources in and near Albany as stated in the following sentence.

P7L19-21: If there was a large difference in the mass fractions of the aerosol components before, during, and after the FW event, this would be interesting to see in pie charts, e.g. in the supplement.

P7L22: If the FW event lasted from 23:00 until 02:00 (line 1, same page), this is 3 hours, not 4 hours.

P7L24: What is "volume mobility diameter"?

P7L27: Not the surface ionization but the slow vaporization of K-containing species will make the K size distributions complicated.

P8L11: Can the authors provide any information on the potential sources of the evening BBOA?

P8L19: It should be Fig. S3b, not S3d.

P8L24 and other locations: Percentages given with a precision of 0.1% and at the same time with likely uncertainties of 10% or more do not make sense.

P8L6-P9L7: This section with the discussion of the PMF results is not very satisfying. The discussion on the background aerosol sources seems very incomplete and short and does only provide a rudimental overview over the various aerosol sources and their variations. Also the discussion on FW effects on the organic aerosol seems rather incomplete. It would be interesting to have a discussion on how fireworks activities like the Independence Day fireworks quantitatively affect the concentrations of the various types of organic aerosol: HOA – by more people being on the streets to observe the display; BBOA – as a by-product of fireworks burning; FW-OA – as a direct emission of the fireworks; OOA – as a potential product of FW-related gas phase emissions. Especially in order to make this study relevant to a broader audience and for FW events outside of Albany, NY this could be very helpful.

P9L13-15: I have the impression that the calculation of organopotassium is highly uncertain. Due to uncertain RIE, slow vaporization effects and surface ionization of potassium the absolute K concentration already is uncertain. The calculation of in-organic potassium is probably even more uncertain due to additional uncertainties of the ammonium fraction of the various salts. Finally, the difference of the two is even more uncertain. If this approach is used, an estimate of these uncertainties would be needed. The fact that resulting organopotassium correlates with FW-OOA is not surprising since potassium correlates well with FW-OOA and organopotassium is probably only a relatively small fraction of total potassium.

P9L15-18: It is unclear to me what the purpose of the calculation of the density of the aerosol is. In this calculation the contribution of refractory aerosol components with

likely high density (metal salts or oxides) and of black carbon is neglected. With these additional uncertainty the resulting uncertainty of the density calculation becomes so large that also a typical density for urban aerosol could be assumed.

P10L4: I do not agree with the assumption that the ratio of FW-related aerosol and background aerosol does not change during transport: While the FW-related aerosol gets diluted during transport, this is not the case for the background aerosol.

P10L6: In addition to the 2-hour average also the 24-hour average of the aerosol should be mentioned here, if these values are compared to the 24-hour threshold value.

P10L7-10: Here the temporal trends of various aerosol components measured during the time after the FW display are used as evidence for conversion processes within the atmosphere. How can the authors be sure that during this time always air masses were probed, which encountered fireworks activities during the night of July 4 and not simply different air masses with different influence of fireworks were probed?

P10L12: Define the "carrier-to-noise ratio" and explain what kind of information this ratio provides.

P10L14: The local time interval is 3 hours long, the UTC interval only 2 hours.

P10L12 – P11L6: This whole section is very confusing to me. There are many inconsistencies between the observations and their interpretation. E.g.: According to the LIDAR weak easterly winds were measured. This is contrary to the wind directions provided in Figure 1. What is correct? Many of the features that are explained in the text are not really visible in Figure 7. What means "high altitude FW transport" – at which altitude was the fireworks burned? There was "high altitude" and "low altitude" aerosol observed in the LIDAR. It is claimed that the low altitude aerosol was the result of subsidence or diffusion of the high altitude aerosol. However, there is no connection visible in the two clusters in the LIDAR signal. Since all the aerosol data are provided for 10 days and the LIDAR signal are only for a couple of hours it is almost impossible

to see all these short-time features in the aerosol data. The 500 m back trajectory passes over the town of Bennington. However, according to Fig. S8 the affected air mass was at an altitude of 1000 m over Bennington. It is unrealistic that this air contained aerosol from the local fireworks in Bennington. It is also unrealistic that the 500 m altitude air was sampled at ground. The aerosol arriving from Bennington (after 40 km of transport) produces a more intense signal in the LIDAR measurement than the aerosol from the much closer Albany fireworks. Is this realistic? This whole section seems to be highly speculative. If it is not it should be much clearer connected to the LIDAR measurements.

P11L8-9: What does "Chemical characterization and source apportionment ... are quantified .." mean?

P11L21-25: This information is not part of the presented study.

Figure 3: Why where the AMS size distributions measured over such a long time interval. This does not allow distinguishing between size distributions dominated by FW events and those of the background aerosol.

Figure 7: This Figure is rather useless for the reader. The features presented in this figure are not explained. The scale has no units and it is not explained what it shows. The wind markers are not visible. It would be nice to present time on this figure consistently with the other graphs.

Figure S3: Diurnal pattern of an individual event (FW-OOA) does not make a lot of sense.

Supplement page 4: The strong dips in SV-OOA and LV-OOA during the main fireworks event and the peak in the residual signal during this time show that PMF does not clearly separate the various types of organic aerosols. This is often observed when individual events occur only during short fractions of the whole time of measurements. A possible approach to improve the PMF separation could be to apply PMF separately

only to a short time interval around the main FW event.

Figure S4a: This figure shows that BC provides a substantial contribution to the overall aerosol. Therefore it cannot be neglected neither in the comparison of total AMS concentrations with those of other instruments nor in the calculation of aerosol density.

Figure S5: After the main FW event some events with very small particles occur. Can these events be explained?

[Figure]

---

## Referee Comment (RC2) · Anonymous Referee #2 · 6 Aug 2018

Paper Summary: This paper reports on the chemical composition of fireworks from the 2017 independence day celebration in Albany, NY. The chemical composition of the fireworks were measured using an Aerodyne type Aerosol Mass Spectrometer. Data from 2 sites was used to investigate transport and other properties of the aerosols in the plumes. Overall, the paper investigates an interesting phenomenon with applications to other celebrations which utilize fireworks, however, the analysis is lacking in a few key areas, and further analysis is warranted before the paper can be published in ACP.

General Comments: The accurate measurement of potassium (K) with an AMS is difficult. Potassium readily ionizes on the 600 C surface of the heater via surface

ionization. Tuning/detuning of the heater bias in the instrument is important to minimize the surface ionization signal in the resulting mass spectrum since it is not quantitative. The difference in noted RIE for K in this work compared to other work is likely due to different tuning of the instruments and enhancements in surface ionization generated K in this work (page 6 lines 9-14). Ion signals resulting from K surface ionization at m/z 39/41 are more broad than other ion signals as a result of the ionization occurring in a spatially separate area of the ionziation region of the AMS compared to EI ionization. The authors are encouraged to look at and include the raw mass spectra signal for these ions in supplementary information (e.g. see supplementary section in Aiken et al. 2009). Given the uncertainty in the K and the lack of details in the paper, the organic potassium discussion and ion balance may need to be removed or updated with additional analysis. On page 3 line 5, the authors state that potassium evaporates and ionizes slowly. This is not always the case, and is dependent on the anion paired with K. $KNO_3$ boils at 400 C, lower than the vaporizer temperature of the AMS, while $K_2SO_4$ boils at nearly 1700 C, so will take much longer to come off of the vaporizer.

The measurements were performed at 2 different sites approximately 9 km apart (Page 3 line 9). Another key difference in sites is the sampling heights which is mentioned, but not discussed in detail. The ASRC site measures at an altitude of 81m, whereas the ACHD site measures at only 8 m above the ground. Is there anyway to clearly identify that the plumes measured at the 2 sites are similar? Could local emissions from people at the ground and commercially available fireworks/sparklers be contributing to the lower sampling height ASRC? As the authors indicate, the lidar data suggests an elevated plume at 400 m, so are the lower altitude sampling locations more likely sampling ground level emissions mixing up, or higher altitude emissions mixing down? This is an important question since the chemical composition may either be for large fireworks or for consumer fireworks which may have significant differences (e.g. sparklers vs rockets etc.).

The PMF analysis of the data is interesting, but the authors provide no details in why

the 5 factor solution was chosen over higher or lower factor solutions. Did the authors investigate any rotation of the data using the fpeak parameters? In the supplementary, the mass spectrum for the HOA spectra has a higher signal from m/z 55 than 57, this may indicate a mixed cooking and traffic factor (see Mohr et al. 2012). Figure 5 indicates that the difference between the FW and the LV-OOA is primarily due to lower m/z 44 (and by frag-table extension m/z 28) and BBOA markers 60 and 73. While this graphic is useful, it may be more helpful to show this as a % change: (LV-OOA-FW-OOA)/LV-OOA * 100% (e.g. Alfarra et al. 2006). Further, given that BBOA is also elevated anytime the the FW-OOA is elevated, could these two factors be "trading mass" to make up for non-static source mass spectra?

Specific Comments: Page 4 line 9: With a 9 meter inlet and a silica diffusion dryer, could the authors comment on the estimated line losses from their sampling inlet?

Figure 7 should remind the reader of the EDT UTC time difference, or better, change the axis to local time (EDT).

References: Aiken, A. C., et al.: Mexico City aerosol analysis during MILAGRO using high resolution aerosol mass spectrometry at the urban supersite (T0) - Part 1: Fine particle composition and organic source apportionment, 9(17), 6633–6653, 2009.

Alfarra, M. R., et al.: A mass spectrometric study of secondary organic aerosols formed from the photooxidation of anthropogenic and biogenic precursors in a reaction chamber, ACP, 6(12), 5279–5293, 2006.

Mohr, C., et al.: Identification and quantification of organic aerosol from cooking and other sources in Barcelona using aerosol mass spectrometer data, ACP, 12(4), 1649–1665, 2012

Ulbrich, I. M., et al.: Interpretation of organic components from Positive Matrix Factorization of aerosol mass spectrometric data, 9(9), 2891–2918, 2009.

---

## Author Comment (AC1) · 14 Sep 2018

**Response to Reviewer 1 comments**

At first thank you so much for your comments, which are all important for our paper and research. Below you find my point-by-point response to your comments, where I first repeat your comment in italic.

**General comments:**

*In order to qualify for ACP and to provide results that can be transferred to other, similar situations, it would be desirable to extend the data analysis beyond the current state. Some examples are:*

*Comparison of the size distributions (total and species-resolved) measured during the Albany fireworks event, during the arrival of the Bennington fireworks aerosol and during the times not affected by fireworks could provide interesting information on the characteristics and temporal evolution of fireworks aerosols. Comparison of the organics mass spectra of the fireworks aerosol during the Albany fireworks event and during arrival of the Bennington fireworks could provide information on aging of fireworks aerosol. For this purpose the fireworks contribution to the organics needs to be determined in a different way, e.g. by separate PMF analysis of both time intervals or by determining the difference between event and non-event times, instead of using a constant PMF factor*

**Thank you for this suggestion. Actually combining the LiDAR measurements, it made more sense that the second enhanced peak was caused by atmospheric mixing which brought the higher aerosol down to the level of the sampler. This is what was measured by the instruments, instead of the transport from Bennington, as we had originally proposed. So we modified this part of the manuscript.**

**To compare the mass spectra, we used data from the second peak hour (July 5 08:00-09:00) and the one of the unaffected previous hour (July 5 06:00-07:00), which shows the relative changes of these mass spectra when the plume that had been aloft injected into the surface level. We did not compare this period to the FW event one, mainly due to the complexity of the organics, which involved the considerable mixing of HOA and BBOA into the organics in addition to the FW-OOA. We did add detailed analysis of this second peak in Section 3.3, which, as suggested, represents the aerosol that aged for a number of hours at a higher level.**

*A more general determination of the contribution of fireworks aerosol to the aerosol burden at a city like Albany, including an estimate on how large the contribution within the affected days or the full year is, in comparison to contributions from other sources would enable the reader to better judge the relevance of such events. This could also include an estimate of the indirect contribution of such an event due to enhanced traffic emissions around the event.*

**Thank you for this suggestion, which would be helpful for people to judge the relevance the FW event. Simply consider the FW display as point source and making some other estimates and assumptions assumption, we roughly estimated the FW aerosol emission based on Gaussian dispersion equation. The result was about 70 kg, and of the same magnitude as the total PM$_{2.5}$ emission from the Albany highway traffic (based on the EPA air emission inventories) in a half day (in Section 3.2.2). This would help give the readers a more direct picture and comparison. At the same time, we compare the HOA concentration around the event to the one around daily rush hour to underline the enhanced emission from traffic (in Section 3.2.1).**

**Detailed comments:**

*Several fireworks events were identified during the measurement period, based on the appearance of the potassium ion in the aerosol mass spectra. I think the authors should explain the criteria more specific here since the potassium ion is also observed with the AMS when measuring biomass burning aerosols. It would be interesting whether during the observation of fireworks events with the AMS also other ions, which could be from the fireworks, were observed, e.g. Sr, Na or other metal ions.*

**Thank you for this comment. After checking high resolution mass spectra, it did show significantly enhanced signals for Rb, but seems to have missed others. We have added this information into the text.**

*I have a general problem with the description of the measurement locations. According to Figure 1, the wind direction was approximately 180-270 during the main FW event. According to Figure S1 the ASRC site was located about NW of the fireworks site, i.e. it needs wind from approx. 100-130 to transport the fireworks aerosol to the measurement site. The ACHD site was located approx. to the SE of the fireworks site, i.e. it needs wind from approx. 280 -320 to transport fireworks aerosol to the measurement site. While for the relatively close ACHD site general dispersion of the fireworks aerosol in the area would probably also transport aerosols to the measurement site, this seems unlikely for the ASRC site, which is almost in the opposite direction compared to the needed wind direction. How can you explain the transport of aerosol from Empire State Plaza to the ASCR site under these meteorological conditions?*

**Really thank you for this comment. During the night of July 4, the wind speed is so low (general below 0.5m/s), that it is classified as calm. We checked additional data from nearby airports and other locations, and all showed calm or no data for the wind direction, which matched the weather map, showing that Albany was controlled by a stagnant high pressure pattern. So we believe the surface wind direction from Voorheesville during the night of July 4$^{th}$ was not representative of actual conditions, namely calm at the surface for the Albany. To reduce confusion, we deleted the wind direction data with wind speed below 0.5 m/s during the night of**

**July 4.**

*I find the presentation of time series of aerosol concentrations (e.g. in Figs. 2, 4, and 6) stacked on top of each other not very helpful when one tries to see temporal variability of any of the mass concentrations. Generally, temporal variability is seen well only for the lowermost trace and in addition – for graphs which show fractional contributions like Fig. 2 – for the uppermost trace. For all other traces only the largest changes can be identified in this type of graph*

**Thank you for this comment. We have modified the these figures to make the variation more clear.**

*The presentation of all the aerosol data for a 10.5 days interval makes it hard to identify the detailed features during the main fireworks event which were discussed within the text. It would be very helpful for the reader to have graphs which show only the time interval around the fireworks event. In addition, it would be very helpful if Figure 7 would show the same time interval as those figures and if the time axes would agree (i.e. not using UTC for one graph and local time for the others).*

**Thank you for this comment. We have added the suggested figure, which only covered the main FW event in section 3.3. At the same time, we modified the time to EDT time in new Figure 9.**

*P2L4: Fireworks aerosol is mainly found in PM1, not in PM2.5, as shown in this manuscript as well as e.g. in the Drewnick et al., 2006 paper.*

**Thank you for this comment. We have corrected this.**

*P2L11-15: Here consumption figures of fireworks in the US are given. It would be interesting to also have numbers for July 4 and may be for the Independence Day fireworks in Albany. The 42% increase of 24-hour PM2.5 – how much is this in absolute concentrations?*

**Thank you for this comment. We added the consumption information for July and the absolute concentration relating to the 42% increase. For the consumption of fireworks in Albany, we cannot find the precise amount, while we estimated based on the shell numbers (10000), which we believe was in the range of 2000-10,000 kg (only firework compounds excluding the shell weight).**

*P2L21: Here two Sun et al. papers are cited. The Sun et al., 2010 paper is not in the reference list. Both papers do not seem to be adequate references for the general measurement capabilities of advanced aerosol mass spectrometry technology like the Jayne et al. and the DeCarlo et al. papers.*

**Thank you for this comment. We deleted the " Sun et al., 2010" one.**

*P3L4-7: Here it seems that some statements are exaggerating. Why was the meteorological situation such that it provided a "unique opportunity" for investigation of the contribution of FW displays to ambient aerosol concentrations? As stated above, the wind direction seemed to be even very unfavorable. Also I do not see how the "health influence" of FW-related aerosols was investigated in this study.*

**Thank you for this comment. We modified this sentence to make it more clear and deleted the exaggerated expressions.**

*P3L16 and L22 and P4L15: It would be helpful if the authors would specify above what the elevation was measured. I guess in L16 it is a.s.l. and in L22 it is above ground, on P4L15 it could be both. Furthermore, if here and in the following paragraph some detailed information is provided, it would be nice if this would be done more consequently: Why is the sampling height provided for the ACHD site but not for the ASRC site? Where was the AMS instrument located, where the SMPS and the pDR-1500? Why is the sampling line length for the inlet duct at ASRC (?) provided, but not the length of the sampling line from the duct to the instruments and also not for the inlet at the ACHD site? For estimate of inlet line losses, also the diameters of the lines and their orientation would be needed.*

**Thank you for this comment. We added "above sea level" into the text, and described the detailed information about the connection between instruments, and calculate the particle loss based on the mentioned software by von der Weiden et al. (2009).**

*P3L19: According to the statement here, the gap in the measurement data was due to maintenance of the instruments. To me it seems very strange that maintenance started at midnight, that maintenance lasted for 44 hours, and that all instruments have exactly the same downtime.*

**Thank you for this comment. At the early morning 0f 07/01, the RH of the sample flow reported by PDR was higher than 40% (and was not corrected until the evening of 07/02), which was too high for consistent sampling by the AMS. For neatness, and being cautious, we flagged the data beginning end at 06/30 23:59, and ending at 07/02 20:00.**

*P5L5 and the whole paragraph: ". . . slow evaporation and ionization of potassium . ..". "evaporation" should be "vaporization". While potassium might vaporize slowly, ionization is probably not slow. In addition to this effect, the quantification of potassium in the AMS is also difficult due to the fact that potassium can be surface-ionized at the vaporizer surface during the vaporization process. This effect will directly affect RIE of potassium. Since this effect  –  and as a consequence the RIE of potassium  –  strongly depends on the vaporizer temperature, the history of the vaporizer and the tuning of the AMS, it makes little sense to use RIE values for potassium obtained in measurements with other AMS but it would be better to determine the own RIE value.*

**Thank you for this particularly helpful comment. We corrected the statement, and we used the RIE for K based on the lab pure $KNO_3$ test, as shown in text. At the same time, we added more detailed information in supplement to discussion about the $RIE_k$ also based on lab pure $K_2SO_4$ test.**

*P6L2: The statement that "all instruments exhibited the same general behavior" seems a bit exaggerated. There are sometimes strong differences between the various PM2.5 measurements up to more than a factor of two. There are also sometimes strong differences in the trends.*

**Thank you for this comment. We deleted this exaggerated expression, and use "exhibited generally similar behavior (that is, high and low excursions)" to make it clear.**

*P6L3-5: I also do not agree with the explanation of the increasing trend in aerosol concentrations with hygroscopic growth. An increasing trend is observed from 6/27 until 6/29 according to Figure 1. During this time RH values varied strongly and no correspondence of the increase in PM and RH can be observed for this whole time interval. Furthermore, hygroscopic growth of particles should not affect AMS measurements of mass concentrations and also not measurements performed with a dryer before the instrument.*

**Thank you for this comment. You are correct. We deleted the sentences about "hygroscopic growth".**

*P6L8-11: The aerosol concentration peaks during the nights of July 2-3 were also identified as FW-related due to their high K signal. Is there any other evidence that these peaks are actually from FW – e.g. similar K/S ratio as observed during the July 4 peak? Was there fireworks reported or observed in the area around the measurement sites? According to the magnitude of these additional FW peaks either a massive fireworks event in the order of 30% of the July 4 fireworks or fireworks which were burned very close to the AMS site would be needed to generate these peaks. Is this realistic?*

**Thank you for this comment. The enhanced Rb ion signal would also support that the nights of July 2-3 was affected by Firework aerosol. From June 30 (even as early as June 24), there were firework displays over New York State (www.newyorkupstate.com/events/2017/06/4th_of_july_ fireworks_in_upstate_ny_list_of_2017_events_celebrations_parades.html), which may contribute the higher K signal at nights of July 2 and July 3. Also during these nights, the neighborhood around the measurements would also burst the consumer fireworks. For the corrected K signals, the K signals of July 2-3 was only about 10% of of July 4. To highlight the key Independence Day event, we modified text to underline the Independence Day FW event, instead of discussing three events.**

*P7L3-5: It is very hard to assign all these numbers to the respective species.*

**Thank you for this comment. We added these numbers to pie charts to aid their clarity.**

*P7L8-10: If SO₄ and NO3 are from K-rich salts and if slow vaporization of K results in under-estimation of K, this should also result in under-estimation of the related SO4 and NO3.*

**We agree with your comments. To correct the SO$_4$ mass concentration, the K/S ratio of 2.75 and estimated K mass concentration was used to estimate the SO4 mass concentration, as show in Supplement. As shown in Supplement, after this correction, the AMS matches SMPS results very well, and the particle mass fraction was near the one reported by Drewnick et al. (2006). However, due to lower boiling point of KNO$_3$ (400 °C) than AMS vaporizer temperature (660 °C), KNO$_3$ was though to be well measured by AMS (Drewnick et al., 2015). So in this study, we only corrected the SO$_4$ mass concentration.**

*P7L10-11: I do not agree with the statement here. According to Figure 2, organics and K show broader peaks than SO4. If all three are from the same source (fireworks), is this difference a sign of chemical processes occurring with the FW aerosol?*

**Thank you for this comment. After careful consideration, we deleted this statement.**

*P7L13-15: If SO4 fraction varies between 8.5 and 16.4% (a factor of two), I would not call this "fairly stable". As a consequence I would not say that this indicates constant sources in and near Albany as stated in the following sentence.*

**Thank you for this comment, and sorry for the former confusion. It was stable for each period separately, instead of for both. We have made this clear.**

*P7L19-21: If there was a large difference in the mass fractions of the aerosol components before, during, and after the FW event, this would be interesting to see in pie charts, e.g. in the supplement.*

**Thank you for this comment. We have added the suggested pie charts.**

*P7L22: If the FW event lasted from 23:00 until 02:00 (line 1, same page), this is 3 hours, not 4 hours.*

**Thank you for this comment. We apologize for this mistake. We have corrected it, and made it clear that the event we consider lasts until until 03:00 LT (EDT).**

*P7L24: What is "volume mobility diameter"?*

**Thank you for this comment. We corrected this to "electrical mobility diameter".**

*P7L27: Not the surface ionization but the slow vaporization of K-containing species will make the K size distributions complicated.*

**Thank you for this comment. We corrected it.**

*P8L11: Can the authors provide any information on the potential sources of the evening BBOA?*

**Thank you for this comment. We added some description into text, and the BBOA was likely from the wood-fire emissions, such as bonfires, barbeques, or other small-scale celebrations.**

*P8L19: It should be Fig. S3b, not S3d.*

**Thank you for this comment. We corrected it.**

*P8L24 and other locations: Percentages given with a precision of 0.1% and at the same time with likely uncertainties of 10% or more do not make sense.*

**Thank you for this comment. We corrected all these values.**

*P8L6-P9L7: This section with the discussion of the PMF results is not very satisfying. The discussion on the background aerosol sources seems very incomplete and short and does only provide a rudimental overview over the various aerosol sources and their variations. Also the discussion on FW effects on the organic aerosol seems rather incomplete. It would be interesting to have a discussion on how fireworks activities like the Independence Day fireworks quantitatively affect the concentrations of the various types of organic aerosol: HOA – by more people being on the streets to observe the display; BBOA – as a by-product of fireworks burning; FW-OA – as a direct emission of the fireworks; OOA – as a potential product of FW-related gas phase emissions. Especially in order to make this study relevant to a broader audience and for FW events outside of Albany, NY this could be very helpful.*

**Thank you for this comment and we agree with you. We have added one paragraph to describe the variation of the different organic aerosols, and compared to the daytime to show the possible sources, including those directly influenced by the FW display events.**

*P9L13-15: I have the impression that the calculation of organopotassium is highly uncertain. Due to uncertain RIE, slow vaporization effects and surface ionization of potassium the absolute K concentration already is uncertain. The calculation of inorganic potassium is probably even more uncertain due to additional uncertainties of the ammonium fraction of the various salts. Finally, the*

*difference of the two is even more uncertain. If this approach is used, an estimate of these uncertainties would be needed. The fact that resulting organopotassium correlates with FW-OOA is not surprising since potassium correlates well with FW-OOA and organopotassium is probably only a relatively small fraction of total potassium.*

**Thank you for this comment, and for helping us realize that organopotassium is not a correct deduction. We corrected the SO₄ data, which lead better attribution for K, and with this better method, there was no K left to attribute to organics. Based on this, we deleted the statement of "organopotassium", and rewrote the former paragraph (moved to section 2.3 as a data analysis part).**

*P9L15-18: It is unclear to me what the purpose of the calculation of the density of the aerosol is. In this calculation the contribution of refractory aerosol components with likely high density (metal salts or oxides) and of black carbon is neglected. With these additional uncertainty the resulting uncertainty of the density calculation becomes so large that also a typical density for urban aerosol could be assumed.*

**Thank you for this comment. We deleted the related statement, and we recalculated the density based on each compound to compare the AMS and SMPS measurement to ensure the RIE for organic and K, as shown in Supplement.**

*P10L4: I do not agree with the assumption that the ratio of FW-related aerosol and background aerosol does not change during transport: While the FW-related aerosol gets diluted during transport, this is not the case for the background aerosol.*

**You are correct. After thinking carefully, we deleted the statement.**

*P10L6: In addition to the 2-hour average also the 24-hour average of the aerosol should be mentioned here, if these values are compared to the 24-hour threshold value.*

**Thank you for this comment. For clarity, we deleted all reference to the 24 hour averaged value.**

*P10L7-10: Here the temporal trends of various aerosol components measured during the time after the FW display are used as evidence for conversion processes within the atmosphere. How can the authors be sure that during this time always air masses were probed, which encountered fireworks activities during the night of July 4 and not simply different air masses with different influence of fireworks were probed?*

**Thank you for this comment. Due to the uncertainty, we deleted this section.**

*P10L12: Define the "carrier-to-noise ratio" and explain what kind of information this ratio provides.*

**Thank you for this comment. We added the related information into the text.**

*P10L14: The local time interval is 3 hours long, the UTC interval only 2 hours.*

**Thank you for this comment. We change the time scale to local time and deleted the UTC interval expression.**

*P10L12 – P11L6: This whole section is very confusing to me. There are many inconsistencies between the observations and their interpretation. E.g.: According to the LIDAR weak easterly winds were measured. This is contrary to the wind directions provided in Figure 1. What is correct? Many of the features that are explained in the text are not really visible in Figure 7. What means "high altitude FW transport" – at which altitude was the fireworks burned? There was "high altitude" and "low altitude" aerosol observed in the LIDAR. It is claimed that the low altitude aerosol was the result of subsidence or diffusion of the high altitude aerosol. However, there is no connection visible in the two clusters in the LIDAR signal. Since all the aerosol data are provided for 10 days and the LIDAR signal are only for a couple of hours it is almost impossible to see all these short-time features in the aerosol data. The 500 m back trajectory passes over the town of Bennington. However, according to Fig. S8 the affected air mass was at an altitude of 1000 m over Bennington. It is unrealistic that this air contained aerosol from the local fireworks in Bennington. It is also unrealistic that the 500 m altitude air was sampled at ground. The aerosol arriving from Bennington (after 40 km of transport) produces a more intense signal in the LIDAR measurement than the aerosol from the much closer Albany fireworks. Is this realistic? This whole section seems to be highly speculative. If it is not it should be much clearer connected to the LIDAR measurements.*

**Thank you so much! We agree that these statements were confusing and unsatisfying. We have rewritten the whole paragraph to make it more clear and solid.**

*P11L8-9: What does "Chemical characterization and source apportionment . . . are quantified .." mean?*

**Sorry for the confusion. We have corrected this sentence.**

*P11L21-25: This information is not part of the presented study.*

**Thank you so much! We write this as one of the conclusion sentences to impress the reader who would care about the FW event. We kept it after little modification.**

*Figure 3: Why where the AMS size distributions measured over such a long time interval. This does not*

*allow distinguishing between size distributions dominated by FW events and those of the background aerosol.*

**We agree. Thank you again. We have re-plotted these, and also included one for background aerosol.**

*Figure 7: This Figure is rather useless for the reader. The features presented in this figure are not explained. The scale has no units and it is not explained what it shows. The wind markers are not visible. It would be nice to present time on this figure consistently with the other graphs.*

**Thank you for this comment. We have re-plotted it.**

*Figure S3: Diurnal pattern of an individual event (FW-OOA) does not make a lot of sense.*

**Thank you for this comment. We deleted it.**

*Supplement page 4: The strong dips in SV-OOA and LV-OOA during the main fireworks event and the peak in the residual signal during this time show that PMF does not clearly separate the various types of organic aerosols. This is often observed when individual events occur only during short fractions of the whole time of measurements. A possible approach to improve the PMF separation could be to apply PMF separately only to a short time interval around the main FW event.*

**Thank you for this comment. We make the case in the supplement that the dip in the SV-OOA can be accounted for and corrected by the residuals. This does not account for the LV-OOA dip, and it does seem there is mixing of these factors during the event. Even with these shortcomings, we kept the long time period PMF in order to show the general character of SV-OOA and LV-OOA.**

*Figure S4a: This figure shows that BC provides a substantial contribution to the overall aerosol. Therefore it cannot be neglected neither in the comparison of total AMS concentrations with those of other instruments nor in the calculation of aerosol density.*

**Thanks for the comment. We add it in the comparison in Fig. S4.**

*Figure S5: After the main FW event some events with very small particles occur. Can these events be explained?*

**Thanks for the comment. It would match the high HOA periods, and be related with the crowd traffic emission. We added the discussion in the last paragraph of section 3.1.**

---

## Author Comment (AC2) · 14 Sep 2018

**Response to Reviewer 2 comments**

At first thank you so much for your comments, which are all important for our paper and research. Below you find my point-by-point response to your comments, where I first repeat your comment in italic.

*The accurate measurement of potassium (K) with an AMS is difficult. Potassium readily ionizes on the 600 C surface of the heater via surface ionization. Tuning/detuning of the heater bias in the instrument is important to minimize the surface ionization signal in the resulting mass spectrum since it is not quantitative. The difference in noted RIE for K in this work compared to other work is likely due todifferent tuning of the instruments and enhancements in surface ionization generated K in this work (page 6 lines 9-14). Ion signals resulting from K surface ionization at m/z 39/41 are more broad than other ion signals as a result of the ionization occurring in a spatially separate area of the ionziation region of the AMS compared to EI ionization. The authors are encouraged to look at and include the raw mass spectra signal for these ions in supplementary information (e.g. see supplementary section in Aiken et al. 2009).*

**We agree with your comments. We added the raw mass spectrum for K in the supplementary information (Fig.S3), and describe the method used to estimate the K mass concentration. At the same time, we used the RIE of K from our lab calibration instead of using the RIE of others (see Section 2.3).**

*Given the uncertainty in the K and the lack of details in the paper, the organic potassium discussion and ion balance may need to be removed or updated with additional analysis.*

**Thank you for your comments. We deleted the organic potassium discussion, and described the method to correct the SO$_4$ mass concentration based on K/S ratio from the Drewnick et al. (2015) paper. The corrected SO$_4$ leads better attribution for K, and does not include any "organopotassium".**

*On page 3 line 5, the authors state that potassium evaporates and ionizes slowly. This is not always the case, and is dependent on the anion paired with K. KNO3 boils at 400 C, lower than the vaporizer temperature of the AMS, while K2SO4 boils at nearly 1700 C, so will take much longer to come off of the vaporizer.*

**Thank you for your comments. We modified the statement to "**The slow vaporization of potassium salt, especially for K$_2$SO$_4$ due to its higher boiling point (1689 °C) than the vaporizer operation temperature (660 °C), made the quantification of K mass concentrations difficult and prone to large uncertainties**", and added the related information in your above statement into text.**

*The measurements were performed at 2 different sites approximately 9 km apart (Page 3 line 9). Another key difference in sites is the sampling heights which is mentioned, but not discussed in detail. The ASRC site measures at an altitude of 81m, whereas the ACHD site measures at only 8 m above the ground. Is there anyway to clearly identify that the plumes measured at the 2 sites are similar? Could local emissions from people at the ground and commercially available fireworks/sparklers be contributing to the lower sampling height ASRC?*

**Thank you for your comments. The corrected PM$_1$ concentration at ASRC was near to the PM$_{2.5}$ concentration at ACHD, which indicated that the instruments at ACHD missed the strongest aerosol plume. This would be caused by the lower altitude (8m) of this site, compared to that of the Empire State Plaza (48m). Due to short distance between these two locations place and weak easterly winds, the aerosol plume did not efficiently diffuse to the lower level. It seems like that the instrument at ACHD just measured the lowest part of the aerosol plume. We added detailed discussion of the transport in the text(Section 3). Due to the laws in the city of Albany, which did not allow people to light fireworks themselves around Empire State Plaza, and considering the long duration of aerosol concentration; and that this was the only big FW display in the vicinity, it is reasonable to believe the aerosol measured from ACHD and ASRC site were both from the plaza FW displays.**

*As the authors indicate, the lidar data suggests an elevated plume at 400 m, so are the lower altitude sampling locations more likely sampling ground level emissions mixing up, or higher altitude emissions mixing down? This is an important question since the chemical composition may either be for large fireworks or for consumer fireworks which may have significant differences (e.g. sparklers vs rockets etc.).*

**Thank you for your comments. We have rewritten the paragraph in section 3.3 to describe the transport process. Our best explanation is that the wind caused different types of transport at different altitudes (or layers in the collapsing nocturnal boundary layer). Above 200m, wind was weak easterlies, while below 200m, the wind was still disarrayed (or calm). It seems that the higher level wind transported the most intense high level firework plume (such as the big rockets) faster to the west, while the lower level firework plume will move slower, which would match the time difference between the occur of these two plumes. We think our instrument measured the lower level aerosol plume during the event period and missed the higher level one. And based on the Albany Firework law, there were no consumer fireworks allowed at Empire State Plaza. So we think these two plumes were both from the FW displays, while the higher part moved faster to west.**

*The PMF analysis of the data is interesting, but the authors provide no details in why the 5 factor solution was chosen over higher or lower factor solutions. Did the authors investigate any rotation of*

*the data using the fpeak parameters?*

**Thank you for your comments. We added the related discussion in the supplement to show why we choose 5 factors. And we did the analysis using different fpeak parameters with a step of 0.2. We choose 0.2 for fpeak value based on the lowest Q.**

*In the supplementary, the mass spectrum for the HOA spectra has a higher signal from m/z 55 than 57, this may indicate a mixed cooking and traffic factor (see Mohr et al. 2012). Figure 5 indicates that the difference between the FW and the LV-OOA is primarily due to lower m/z 44 (and by frag-table extension m/z 28) and BBOA markers 60 and 73. While this graphic is useful, it may be more helpful to show this as a % change: (LV-OOAFW- OOA)/LV-OOA \* 100% (e.g. Alfarra et al. 2006). Further, given that BBOA is also elevated anytime the FW-OOA is elevated, could these two factors be "trading mass" to make up for non-static source mass spectra?*

**Thank you for your comments. We added the related information about the signal of m/z 55 and 57 into the text, and relative variation of the mass spectrum of FW-OOA and LV-OOA in the supplement. There is also information about the temporal behavior of the factors, indicating some difference between the BBOA and FW-OOA. There could be some mixing of these factors during the event, but this would change the FW-OOA by at most roughly 10%.**

*Specific Comments: Page 4 line 9: With a 9 meter inlet and a silica diffusion dryer, could the authors comment on the estimated line losses from their sampling inlet?*

**Thank you for your comments. We calculated the particle loss based the particle loss calculator (http://www.mpch-mainz.mpg.de/~drewnick/PLC/). The flow rate for the 9 meter inlet was roughly a few hundred LPM, so we found there was little loss for it. We calculated the particle loss for the other tubes and dryers.**

*Figure 7 should remind the reader of the EDT UTC time difference, or better, change the axis to local time (EDT).*

**Thank you for your comments. We corrected to EDT time scale.**

*References: Aiken, A. C., et al.: Mexico City aerosol analysis during MILAGRO using high resolution aerosol mass spectrometry at the urban supersite (T0) - Part 1: Fine particle composition and organic source apportionment, 9(17), 6633–6653, 2009.*

*Alfarra, M. R., et al.: A mass spectrometric study of secondary organic aerosols formed from the photooxidation of anthropogenic and biogenic precursors in a reaction chamber, ACP, 6(12), 5279–5293, 2006.*

*Mohr, C., et al.: Identification and quantification of organic aerosol from cooking and other sources in Barcelona using aerosol mass spectrometer data, ACP, 12(4), 1649 – 1665, 2012*

*Ulbrich, I. M., et al.: Interpretation of organic components from Positive Matrix Factorization of aerosol mass spectrometric data, 9(9), 2891 – 2918, 2009.*